# Intra-gastric phytoliths provide evidence for folivory in basal avialans of the Early Cretaceous Jehol Biota

Yan Wu[1], Yong Ge [1,2], Han Hu[3], Thomas A. Stidham [1,4], Zhiheng Li [1]✉, Alida M. Bailleul [1] & Zhonghe Zhou[1,4]

Angiosperms became the dominant plant group in early to middle Cretaceous terrestrial ecosystems, coincident with the timing of the earliest pulse of bird diversification. While living birds and angiosperms exhibit strong interactions across pollination/nectivory, seed dispersal/frugivory, and folivory, documentation of the evolutionary origins and construction of that ecological complexity remains scarce in the Mesozoic. Through the first study of preserved in situ dietary derived phytoliths in a nearly complete skeleton of the early diverging avialan clade Jeholornithidae, we provide direct dietary evidence that *Jeholornis* consumed leaves likely from the magnoliid angiosperm clade, and these results lend further support for early ecological connections among the earliest birds and angiosperms. The broad diet of the early diverging avialan *Jeholornis* including at least fruits and leaves marks a clear transition in the early evolution of birds in the establishment of an arboreal (angiosperm) herbivore niche in the Early Cretaceous occupied largely by birds today. Morphometric reanalysis of the lower jaw of *Jeholornis* further supports a generalized morphology shared with other herbivorous birds, including an extant avian folivore, the hoatzin.

Angiosperms are primary components of the Earth's terrestrial biomes today, structuring and sustaining fundamental aspects of global ecosystems. While great progress has been made in understanding the origin and early diversification of angiosperms—commonly referred to as Darwin's "abominable mystery"—over the last two decades[1–10], their early fossil record tells an incomplete story, particularly with regards to the origins and assembly of these initial angiosperm supported food webs. Birds play perhaps an oversized ecological part in modern angiosperm-dominated settings with their multifarious roles as herbivores, pollinators, and seed dispersers of flowering plants, but the early avialan fossil record poorly documents the evolution of these important aspects of their ecology. However, the Early Cretaceous Jehol Biota of Northeastern China preserves a forested habitat comprising some of the oldest angiosperms like *Archaefructus*[11,12] along with the oldest known diverse assemblage of birds. Given the great diversity of skull shapes, dentitions, and other morphological variation present among these early diverging Chinese avialans that were largely arboreal in their habits[13–16], they are thought to represent a similarly broad range of diets, including insects, vertebrates, and plant components such as fruits, with some direct fossil evidence[15,16] and indirect evidence such as the presence of clusters of gizzard stones interpreted as gastroliths. However, that breadth falls short of the extraordinary range present among extant birds. Given the current strong ecological links among birds and angiosperms and fossil indications of past connections near their origins, the rise of angiosperms might inextricably be linked to the rise of birds during the Cretaceous.

[1]Key Laboratory of Vertebrate Evolution and Human Origins of Chinese Academy of Sciences, Institute of Vertebrate Paleontology and Paleoanthropology, Chinese Academy of Sciences, 142 Xi-zhi-men-wai Street, 100044 Beijing, China. [2]Department of Archaeology and Anthropology, University of Chinese Academy of Sciences, 100049 Beijing, China. [3]Department of Earth Sciences, University of Oxford, Oxford OX1 3AN, UK. [4]College of Earth and Planetary Sciences, University of Chinese Academy of Sciences, 100049 Beijing, China. ✉e-mail: lizhiheng@ivpp.ac.cn

To further document ancient interactions among angiosperms and birds, we have applied a methodology to explore the reconstruction of the herbivorous diets of fossil avialans in the hopes of making progress in delineating early bird-angiosperm interactions. Here, we expand the existing direct evidence related to early diverging avialan feeding habits by sampling materials from the digestive tract (locating the stomach region) of a subadult individual of the long bony-tailed early diverging taxon *Jeholornis*. This application of phytolith analysis of digestive tract materials in fossil avialans reveals that they were likely comprised of leaves from the magnoliid angiosperm clade. This direct dietary evidence in *Jeholornis* expands the known avialan herbivorous diet from angiosperm fruits to include angiosperm leaf-eating (folivory) in the Cretaceous forested environment[17,18]. Furthermore, a reinvestigation of the morphometric analysis and the qualitative comparisons of the lower jaw morphology of *Jeholornis* reveals particular similarities with the hoatzin (a crown avian folivore),

as well as other more generalized plant-consuming birds. Given the phylogenetic placement of *Jeholornis* near the origin of Avialae and its demonstrated herbivorous diet, avialans likely shifted their diet and ecology early in their evolution away from that of their closely related predatory Mesozoic outgroups like dromaeosaurs and troodontids.

As the earliest diverging avialan lineage besides *Archaeopteryx*[19] and one of the best-known early diverging avialans, the arboreal *Jeholornis prima* and other closely related species within the clade Jeholornithidae are represented by over a hundred well-preserved complete specimens mainly recovered from the ~120 Ma Jiufotang Formation of the Jehol Biota[20]. This iconic long bony-tailed species (Fig. 1) is a classic exemplar of early evolution in avialan phylogeny, characterized by a largely reduced dentition, a plesiomorphic akinetic diapsid skull[21], and an exceptionally elongated bony tail decorated with fan-shaped rectrices[22]. Recently, geometric morphometric analyses of the mandible and three-dimensional (3D) comparisons with the

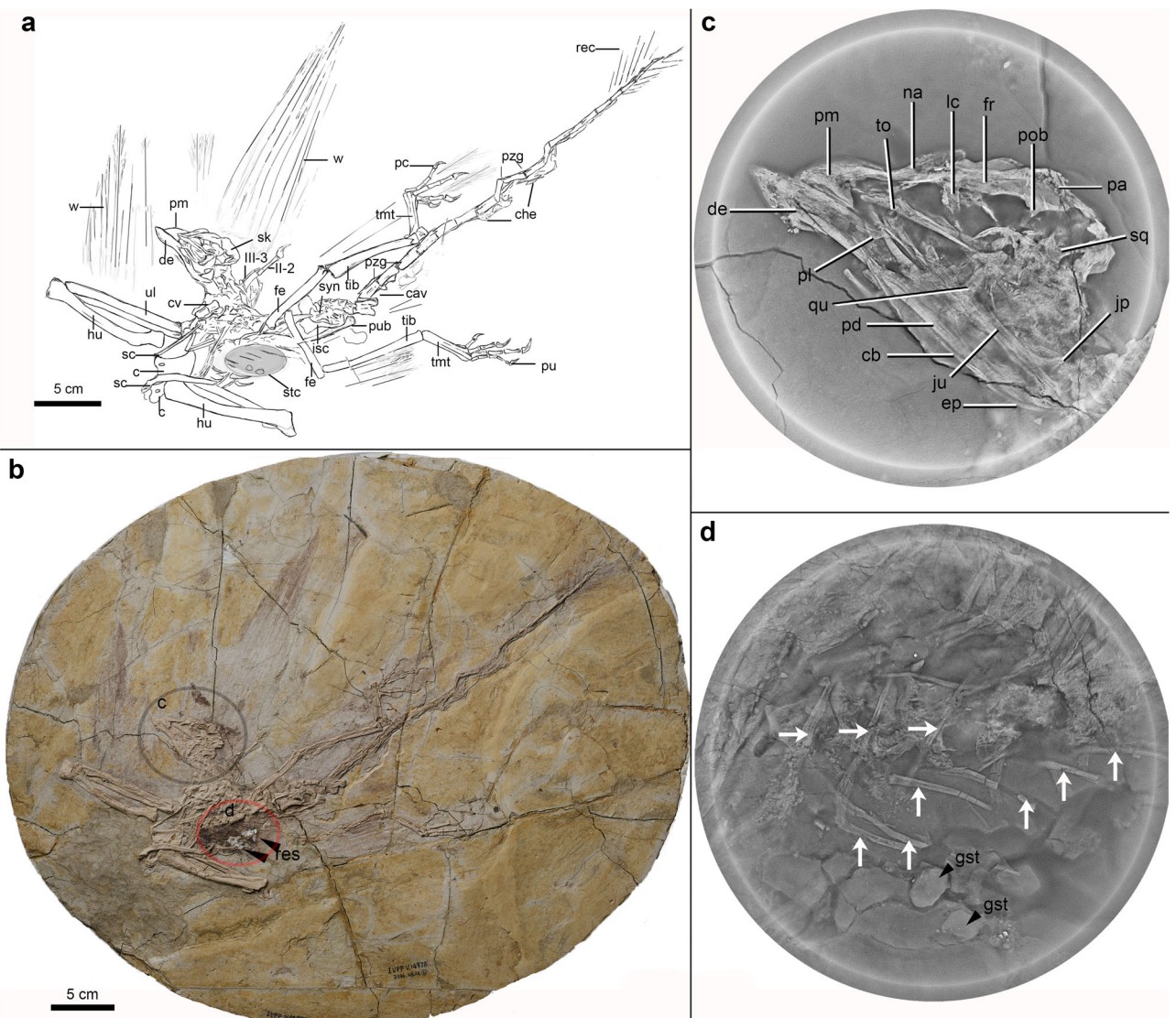

**Fig. 1 | Line-drawing, photography, and CL scans of the specimen of *Jeholornis prima* (IVPP V14978).** The stomach area from which the phytoliths were extracted is shaded gray in (**a**) and shown in the CL scan in (**d**). **c**, **d** are focused views of the skull and thoracic region from the CL scans. Black arrows in (**d**) indicate the gastroliths preserved associated with the residue (black arrows in **b**) sampled from the digestive tract, and the parallel white arrows indicate thoracic and sternal ribs. c coracoid, cav caudal vertebrae, cb ceratobranchial, che chevron, cv cervical vertebrate, de dentary, ep epibranchial, fe femur, fr frontal, gst gastroliths, hu humerus, II-2 manual digit phalanx II-2, III-3 manual digit phalanx III-3, isc ischium, jp jugal process, ju jugal, lc lacrimal, na nasal, pa parietal, pd postdentary, pl palatine, pm premaxilla, pob postorbital, pu pedal ungual, pub pubis, pzg pre- and post-zygapophyses, qu quadrate, rec rectrix, res residue, sc scapula, sk skull, stc stomach content, sq squamosal, syn synsacrum, tib tibiotarsus, tmt tarsometatarsus, to tooth, ul ulnae, w wing feathers.

alimentary tract contexts of extant birds indicate that angiosperm fruits were components of *Jeholornis*' diet[17], possibly linking the foraging of one of the earliest birds to the reproductive habits of early flowering plants.

Plant phytoliths are composed of opaline silica deposited within and between cells formed by monosillicic acid brought into the plants through the uptake of water[23]. Phytoliths have a long fossil record to near the origin of plants in the Paleozoic, are resistant to dissolution (particularly in highly oxidized depositional settings)[24], and are of particular use in paleontological and archeological disciplines because of their highly diagnostic utility in plant taxonomy[23]. In addition to their archeological use, phytoliths have been used to reconstruct the diet of dinosaurs through analysis of their feces and dentition related residues[25,26], providing direct evidence of dinosaur-plant interactions. Here, we show successful deep-time application of dietary phytolith analysis from residues of the digestive tract in an early diverging bird fossil.

## Results

### Characterization of ancient phytoliths

The subadult specimen of Jeholornithidae (IVPP V14978) that we sampled was excavated in western Liaoning Province near Chaoyang City in northeastern China, and it is one of the smallest individuals known from this clade (Fig. 1, Supplementary Figs. 1 and 2, and Supplementary Table 1). The new specimen is taxonomically referred to *Jeholornis prima* based on the presence of a long bony tail, a strut-like coracoid, and a largely edentulous jaw (see detailed description and comparisons in Supplementary Information Note 1). We successfully obtained phytoliths from fossilized digestive tract residues (in particular, deriving from the gastric region) within the thoracic rib-cage of this new specimen (Fig. 1 and Supplementary Information Note 2). Careful processing of samples extracted from within the bird specimen produced a total of 418 phytoliths (totaling ~2 g) (Fig. 2 and Supplementary Fig. 3). Characteristics of these phytoliths include a dull color under orthogonal polarized light, and they have the same optical properties as opal under transmitted light microscopy. SEM−EDS

analyses demonstrate that the main component present in the fossil phytoliths is silicon dioxide, typical of phytoliths (Supplementary Fig. 4). Importantly, two samples of the sedimentary matrix surrounding and external to the fossil skeleton yielded negative results, and no phytoliths were recovered. Fossil phytoliths derive from only within the body wall of this avalian individual. This result further supports our initial identification of the residues as digestive tract content within the bird and demonstrates that the phytoliths are endogenous to the bird, not the result of contamination from the adjacent (lacustrine) sediments.

### Phytolith identification and extant taxa comparisons

These fossil phytoliths were identified through comparison with more than 4000 kinds of modern plant phytoliths using databases of current international phytolith nomenclature, published literature[23,27-44], and our work. Our comparisons demonstrate that most of the phytoliths (284, about 68% of the total sample) are characterized by a blocky body with wavy ridge ornamentations and a size of ranging between ~45 and 90 µm.

Initial comparisons of this more common grouping indicate that the most likely identifications of the fossils are among five types of modern phytoliths which similarly exhibit a blocky shape and ridgeline ornamentation, including those derived from the aerial parts of ferns, the leaves of conifers, the bulliform cells of Poaceae (grass) leaves, and the leaves of common broad-leaved trees and Magnoliales (Supplementary Fig. 5). When the fossil phytoliths were compared in detail to these five categories of modern phytoliths, the strongest morphological similarities are to the phytoliths from the extant *Lirianthe coco* (Magnoliaceae) and *Machilus nanmu* (Lauraceae) leaves (Supplementary Fig. 5f-j). All phytoliths have a blocky body with polygonal margins and a polyhedral shape decorated with strong wavy ridgelines. An extended comparison with other species from Magnoliaceae helps to support this result (Supplementary Figs. 5 and 6). The blocky bodies from ferns (Supplementary Fig. 5a), conifers (Supplementary Fig. 5b), and Poaceae (Supplementary Fig. 5c, d) are not polyhedral, but rectangular. In addition, the shape of the phytoliths from common broad-

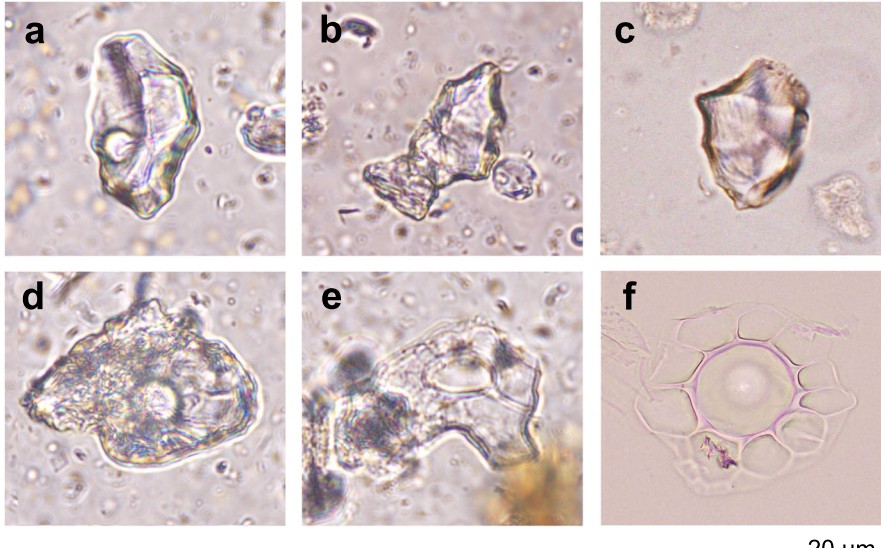

**Fig. 2 | Comparison between fossil phytoliths extracted from the digestive tract of *Jeholornis prima* (IVPP V14978) and modern phytoliths. a, b** Fossil blocky phytoliths with wavy ridgelines from the stomach content of *Jeholornis prima* (IVPP V14978), consistent with the blocky phytoliths in modern magnoliid leaves; **c** blocky phytoliths with wavy ridgelines, extracted from *Lirianthe coco* leaves (an extant species of Magnoliales) collected from Guangxi Province, China; **d, e** fossil phytoliths with radiate lines and a conical projection in the center, which is similar to the hair base phytoliths in modern plants; **f** hair base phytoliths extracted from extant *Ficus tikoua* leaves collected from Gongga Mountain, Sichuan, China. Scale bar is equal across all panels.

leaved trees decorated with the weak ridgeline ornament (Supplementary Fig. 5e) do not resemble the fossil phytoliths. As the sister group of Magnoliales, Laurales may share several morphological characters because of their common ancestry. Since the fossil phytoliths are nearly identical in their morphology with the leaf phytoliths of extant Magnoliaceae and Lauraceae (belonging to Magnoliales and Laurales), we conclude that these blocky phytoliths most likely derive from the leaves of magnoliids.

Another type of fossil phytoliths (*n* = 2) with a round shape and a protuberant center (Fig. 2d, e) is present in the sampled digestive tract residue, and its two key characteristics (Supplementary Fig. 7) resemble phytoliths in modern eudicot leaves (Fig. 2f). One hundred and thirty-two phytoliths (accounting for 31% of the total assemblage) from the fossil sample are not identified to a particular plant taxon because they do not match any known morphotype from our own and other published references (Supplementary Fig. 3k). These unknown phytolith morphotypes suggest that many ancient plant species of the Early Cretaceous Jehol Biota await discovery, and that the diet of *Jeholornis* likely incorporated more plant species than just those of magnoliids.

## Cretaceous emergence of magnoliids

The most recent dates estimated for the divergence within crown Magnoliales and Laurales are 130 Ma (95% HPD: 119–142 Ma) and 131 Ma (120–144 Ma) in the Early Cretaceous based on molecular clock dating, with a comprehensive adjustment derived from fossil calibrations[45]. Other estimated dates include 117 Ma (115–121 Ma) and 107 Ma (105–112 Ma) based on a mega-phylogeny of plastid genomes[8], and 119 Ma (107–131 Ma) and 117 Ma (106–130 Ma) according to a reconstructed time frame derived from transcriptomes and nuclear genomes[46]. Those estimates are consistent with the age of the Jiufotang Formation (and the *Jeholornis* fossils described here) with its dated volcanic ash layer of ~120 Ma[47]. In addition, palynological records also demonstrate the presence of angiosperm taxa in the Jehol Biota, including *Magnoliapollis, Liliacidites, Asteropollis, Knemapollis*, and *Chloranthus*[48]. Our phytolith data indicate that magnoliids comprised part of the dietary resources for *Jeholornis*. Since most of the fossil phytoliths show typical characteristics shared with modern leaves, it is reasonable to hypothesize that leaves (of magnoliids) accounted for at least a modest proportion of the diet in *Jeholornis*.

Clearly, the forests of the Jehol Biota with their diverse communities of gymnosperms and angiosperms provided significant habitat breadth, as well as various abundant dietary items, to its invertebrate and vertebrate inhabitants. Many extant birds occupy a key position as arboreal specialists who can access a wide variety of (forest) resources. The early arboreal bird *Jeholornis* exploited those resources with its demonstrated mixed herbivorous diet[15], including the oldest record of leaf-eating among avialans. Its consumption of fruit, seeds, and leaves may have aided *Jeholornis* (and other early birds) in avoiding competition with contemporaneous non-avialan dinosaurs. *Jeholornis* marks not only a dietary shift among early diverging avialans away from the predatory habits of the close avialan outgroups of dromaeosaurs and troodontids, but also the initial establishment of the arboreal herbivore niche among birds. That arboreal volant herbivory role is present among not only *Jeholornis*, but also early pygostylians such as *Sapeornis, Eogranivora*, and *Hongshanornis* from the same time and region in China[15,16] pointing to a thriving community of ecologically similar Early Cretaceous birds.

## Discussion

This ecological and dietary shift to arboreal herbivory likely impacted the overall early evolution of birds and their adaptation to Cretaceous forests with the rising prominence of angiosperms. One impact in the change in avialan diet from its predatory origins can be seen readily in the occurrence of gastroliths in the herbivorous early diverging avialan

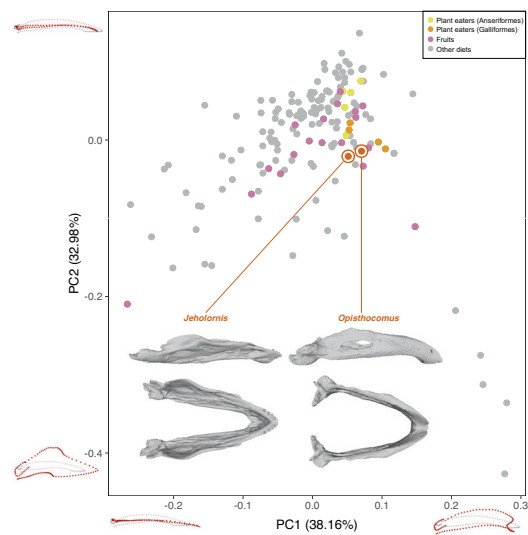

**Fig. 3 | PCA plot of 3D lower jaw shape including *Jeholornis* and extant birds, with a focus on comparison with extant herbivorous birds.** The 3D model of the mandible of *Jeholornis prima* is based on *Jeholornis prima* STM 3–8 (see ref. [17]).

taxa, at least seasonally[15]. In addition, changes to the skull have been recognized during the dinosaur–bird transition, such as the reduction in dentition, the thinning of dental enamel in *Jeholornis*[49], and changes in skull shape[17,21]. Therefore, we reexamined the results of the prior 3D geometric morphometric (GMM) analysis of the mandibles including *Jeholornis* and extant birds[17], to provide supplementary tests to the phytolith analyses regarding leaf consumption by *Jeholornis*. The principal components analysis (PCA) places *Jeholornis* in the center of the overall mandibular morphospace (Fig. 3), indicating that its mandible likely belongs to a generalized functional type, but with potential to evolve in different directions towards more specialized dietary morphotypes, except for some specific diets, like seed crackers[16]. *Jeholornis* also plots very close to other generalized plant consumers such as large number of members of the Anseriformes and Galliformes, particularly along PC1 axis. In addition, it is notable that *Jeholornis* plots very close, along both PC1 and PC2 axes, to the hoatzin, an extant species of specialized folivore. A qualitative comparison of the 3D morphology of the mandibles of *Jeholornis* and the hoatzin also indicates strong similarities, such as the reflexed rostral part of the mandible (Fig. 3). Intriguingly, some other ecologically disparate taxa plot close to *Jeholornis*, including some raptorial birds (*Micrastur* and *Vultur*), which may be the result of convergent evolution in jaw function for tearing food items. The previous interpretation derived from the published mandible GMM analysis focused only on excluding the possibility of granivory in *Jeholornis*[17]. However, our reexamination of those data indicates the potential for a more varied plant-based diet in *Jeholornis*, while also providing an assessment independent of the phytolith analysis supporting leaf consumption as at least part of its diet (Fig. 4). Leaves are typically a lower quality source of dietary resources as compared to other plant parts like fruits and seeds, and concomitantly, a specialized leaf-eating diet is rare among extant birds because leaves alone probably cannot provide the requirements to fuel long sustained flights[50]. As has been proposed for the seasonally available fruits found in *Jeholornis* fossils and the variably present gastroliths, angiosperm leaves likely formed only part of a broad, dominantly herbivorous diet, similar to the role they play in living birds such as mousebirds (Coliiformes).

Our discovery reinforces knowledge of the early ecological linkages among birds and angiosperms (ornithophily) from fruit/seed consumption (and possible dispersal) across multiple avialan lineages (including one of the most early diverging lineages), to the utilization

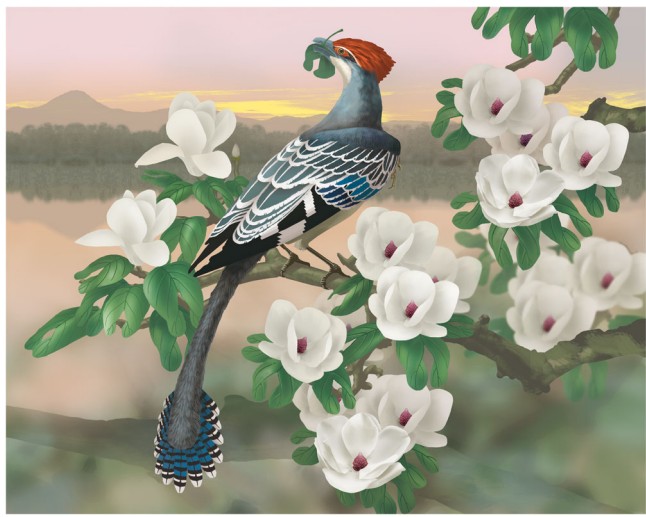

**Fig. 4 | Artist's reconstruction of *Jeholornis prima* with potential angiosperm arboreal herbivorous feeding ecology proposed here.** The leaf of Cretaceous angiosperm was reconstructed as the diet of *Jeholornis*.

of the most abundant and common plant parts like leaves for dietary purposes. In addition, access to plant-derived chemicals such as carotenoids (through dietary consumption) may have had secondary evolutionary effects among birds, where they contribute to the bright coloration of many extant birds.

Palynological records from the Jehol Biota are poorly known[51], and our study with an exceptionally preserved fossil bird provides a new avenue for revealing plant diversity and evolution in a specific region, particularly offering insights into the taxonomy and distribution of early angiosperms in the Cretaceous. In particular, we have documented some of the oldest records of magnoliids, and uncovered previously undocumented plant diversity within the Early Cretaceous Jehol Biota.

## Methods
### Extraction of fossil and extant samples
To obtain the samples for analyses, we scraped both the carbonized sample (potential food residue within the gastric region of the bird), and the surrounding sediments using a new clean razor blade (Fig. 1 and Supplementary Fig. 1; with two gastric samples labeled in the Supplementary Fig. 1). The process of phytolith extraction followed methods slightly modified from refs. 23,29.

Destructive sampling of gastric content followed the requirements and guidelines of the Institute of Vertebrate Paleontology and Paleoanthropology. The permissions were acquired and granted in April 2019 by the collection manager (Binghe Geng), and the institute director (Tao Deng).

Approximately two grams of targeted fossilized stomach material were taken independently on two occasions for phytolith extraction. Initially, approximately one gram of the targeted material was taken for phytolith extraction. The samples were dried, ground into powder, and the resulting dry residue was treated with 10% hydrochloric acid (mass concentration) to eliminate carbonates. Subsequently, 30% hydrogen peroxide was added to each sample to remove organic matter. To concentrate the phytoliths, heavy liquid flotation was conducted using a $ZnBr_2$ solution with a density of 2.3 g/cm³. The heavy liquid flotation process was repeated to ensure sufficient concentration. The residue was washed with distilled water via centrifugation at 1690 × g for 10 min. For further examination, the residue was observed and photographed under a Nikon Eclipse LV100P0L microscope (×500) using Canada Balsam. To obtain a higher quantity of extracted fossil phytoliths, we repeated the aforementioned experiments and collected approximately another one-gram sample from an adjacent region of the fossilized

stomach material. The repeated experiment yielded similar phytoliths and exhibited consistency with the initial one.

Selected leaf samples for extant phytoliths were processed following ref. 52 and detailed procedures are described below: weigh one-gram sample on a balance and place the weighed sample into a clean crucible; put the sample into a muffle furnace and incinerate at 500 °C for 12 h; transfer the obtained ashes to a centrifuge tube; add 6 ml of 10% HCl and shake, allow it to settle for 5 min; add distilled water to the centrifuge tube to reach 10 ml, close the lid tightly and shake; centrifuge at 1690 × g for 8 min; pour off the upper liquid layer; repeat the last three steps until the surface liquid is clear; air-dry the sample in its natural state and then weigh and record the weight of the obtained samples.

### Morphology and chemical characteristics of phytoliths analyzed using SEM−EDS
A phenom Pro X scanning electron microscope (SEM) with EDS was used to examine and document the detailed morphology and chemical composition of these samples. The samples were coated with gold by high vacuum evaporation. The operating voltage of EDS analysis was 15 kv and with the doweling duration of 100−360 s. The main element present in the fossil phytoliths was silicon, according to the EDS results (Supplementary Fig. 3).

### Morphometric data and analysis
The three-dimensional mandibular landmark dataset is adapted from ref. 17. The plots of the *Jeholornis* mandible were compared with those of living birds (Fig. 3). GMM reanalysis were conducted using the function of gpagen() for Generalized Procrustes Analysis, gm.prcomp() for PCA, and plotRefToTarget() for shape variations along PC axes from package "geomorph" in R[53].

The diets were divided into four categories. For generalized plant eaters, plant material comprises >50% of their total diets based on a previous survey. Those generalized plant eaters were divided further into (1) anseriformes, and (2) galliformes based on the great difference in their foraging techniques; (3) fruits; and (4) other diets. The sample list with the adopted diet categories is provided in Supplementary Table 2.

### Reporting summary
Further information on research design is available in the Nature Portfolio Reporting Summary linked to this article.

## Data availability
The authors declare that all data supporting the results of this study are available within the paper and its supplementary information files. All of the specimens used in the GMM part have been published before, including the key specimen *Jeholornis* STM 3−8[16], whose computed tomography scanning slices and segmented STL files are available in MorphoSource (https://www.morphosource.org/projects/0000C1212)[16]. A custom-built Computed Laminography (CL) system at the IVPP was used to image the new specimen, with a focus on the skull and gastric region, and the CL datasets (Supplementary Data 1) are accessible through a publicly accessible framework (https://osf.io/hbxfg/) as compressed tiff images stack data.

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

## Acknowledgements

This research was mainly supported by the National Science Foundation of China (Grant No. 42288201 to Z.Z., Craton destruction and terrestrial life evolution) and NSFC 41877427 to W.Y. and the Strategic Priority Research Program of the Chinese Academy of Science (No. XDB26000000 to Z.L.), the National Key R&D Program of China (2022YFF0801500), and the Major Program of National Social Science Foundation of China (22 and ZD246) also partially supported the work. This research is part of a project that has received additional funding from the European Union's Horizon 2020 research and innovation program under the Marie Skłodowska-Curie grant agreement (No. 101024572 to H.H.). We are grateful for the discussion with Yang Tuo about extant plant taxonomic terminology. Liu Xinzheng, Wen Xiaolei, and Dai Anke helped with sample extraction, and Gao Wei, Wang Jingyi, and Luo Wugan helped with photography. We thank Xu Yong for producing the attractive artistic reconstruction of the new specimen.

## Author contributions

Z.L., Y.W., and Z.Z. designed the study. Y.W., Y.G., H.H., Z.Z., A.M.B., and Z.L. collected the data. Y.W. performed ancient phytoliths experiment and analyzed the phytoliths; Y.W. and Y.G. performed extant phytolith experiment and comparison of phytoliths and Z.L, Y.W., T.A.S., H.H., and Y.G. wrote the paper with inputs from all authors.

## Competing interests

The authors declare no competing interests.
