## [Peer Review File · Nature Communications]

Intra-gastric phytoliths provide evidence for folivory in basal avialans of the Early Cretaceous Jehol BiotaREVIEWER COMMENTS

Reviewer #1 (Remarks to the Author):

In their manuscript, Wu et al. report a new specimen of the basally-diverging avialan Jeholornis that preserves dietary gut contents. The authors demonstrate that those contents include phytoliths, a rugged silica-based structure produced by plants. Based on comparison of SEM imagery of those phytoliths to corresponding structures from a sample of extant plants, the authors argue that they were most likely produced by an angiosperm plant (specifically magnolias and relatives). The authors use these results in combination with a GMM analysis of the mandible preserved in the new specimen to argue that angiosperm-based folivory constituted a significant proportion of the diet in Jeholornis.

This study provides a crucial glimpse into the early co-evolution of angiosperm diversity and avialan feeding ecology, and sheds light on the community structure of the extremely important Cretaceous Jehol Biota. Generally, I find the authors' arguments to be reasonable, compelling and well-supported by the presented data. The artistic reconstruction of Jeholornis (Fig. 4) is exceptional – please pass my regards on to the artist. My only major comments are requests for additional information in some areas to better support their interpretations.

1. The authors do not explicitly discuss – and therefore do not reject – any alternative interpretations of the structures they refer to as phytoliths. The comparative imagery the authors provide is quite convincing that they are, but some justification for the rejection of alternative interpretations should be provided. Or, if there are no other structures they might be, that should be stated explicitly.
2. Related to my previous point, additional comparative information should be provided to the diagnosis of the phytoliths as coming from Magnoliales. Minimally, images of the fossil phytoliths should be added to Supplementary Fig. 4-5 to help readers make comparisons more easily. Also, while I am convinced by the authors' assignments of some pictured phytoliths to Magnoliales (e.g. Fig. 1a-b), I am not convinced by the identification of others as hair base phytoliths. Specifically, to my (admittedly untrained) eye, the phytoliths pictured in Fig. 1d-e looking nothing like Fig. 1f. I recommend either the authors provide more justification for that assignment or drop that line of reasoning completely. I am inclined to suggest the latter – the authors' arguments for angiosperm-focused folivory in the new specimen is convincing enough without the hair cell element.
3. I don't find the GMM aspect of this study very useful or relevant – which I suspect is why little attention is paid to it in the manuscript. As the authors note, it seems to do little beyond suggesting a "generalized dietary morphotype," of which folivory may or may not be a part. The authors' emphasis on similarity with Hoatzin seems unjustified – indeed, there seem to be several taxa in Fig. 3 that are apparently not herbivorous (they are not colored yellow) but are more closely positioned in parameter space to both Jeholornis and Hoatzin than each other. If the authors decide to keep this section, I think it warrants much more discussion: what are those taxa that are close to Jeholornis or Hoatzin that are

not herbivorous? The authors briefly mention that other authors used GMM approaches to reject a granivorous diet for *Jeholornis* – how are the results of the current paper different from those previous studies – or are they?

4. The systematic paleontology section in the supplement should more explicitly justify referral of the specimen to *Jeholornis prima*. I have no doubt that the authors are correct in their referral, but they should discuss not only how the new specimen is like *J. prima* (which they do), but also how they reject other possible taxonomic assignments. Also, the authors should provide some information on the locality from which it was recovered.

5. Related to my previous point, the line drawing (Fig. 1a) should provide a more direct, literal interpretation of the preserved elements. As presented, the “sketch”-like approach is attractive, but not very useful for discerning much meaningful anatomy, especially in the context of taxonomic diagnosis. I urge the authors to use a program like Illustrator to use well-defined lines with consistent line weights and to represent as much discernable anatomy as possible.

6. Based on the data availability statement in the paper, it is not clear to me if the authors have already done this, but I *strongly* urge them to place the digital data associated with this paper (e.g., the CT scan data) on an online repository (e.g., MorphoSource) and provide a link to those data in the final version of the paper.

Some minor note:

Line 61 – some awkward grammar and typos here – try something like: “from the approximate location of the stomach region”

Line 68-69 – “...from their predatory close outgroups...” – it is unclear what the authors are describing here – are they referring to a folivorous diet as a derived feature relative to the presumably ancestral carnivorous diet of close relatives?

Line 69-70 – This should be reworded to make clear that the morphometric analysis being referred to here was done within this same paper.

Line 74 – “over one hundred” rather than “over a hundred”

Line 86 – “high” is misspelled

Line 100 – “careful” is misspelled but also I don’t think you need this word here

Line 106 – extra space before period

Line 120 – what do you mean by Pterydophita? I am by no means well versed in plant phylogenetics, but it is my understanding that the grouping referred to by “Pterydophita” is not monophyletic, and so it is an obsolete word. But I may be wrong!

Line 185 – “few energy” is awkward phrasing – presumably you mean less energy than an alternative food source. I suggest you make that, as well as what those alternatives are, explicit.

Reviewer #2 (Remarks to the Author):

This is a very interesting paper. It deals with time periods outside of my expertise but I nevertheless can comment on the phytolith record. The phytolith record is well conceived, identified, and interpreted. The methods used to isolate and identify phytoliths from the bird's stomach are sound. Magnoliaceae and other basal angiosperm phytoliths such as Lauraceae are well-studied by others and the authors, and it does appear that the bird was eating members of the Magnoliales, supporting other conceptual and interpretive aspects of the paper. How interesting that the diet of a bird this old and important can be studied in this way.

Other parts of the paper are well-written and easy to understand for those like myself who study later time periods in the past. I don't have any substantive comments for revision. I think the paper should definitely be published.

Dolores Piperno

Reviewer #3 (Remarks to the Author):

The work suggests that basal avialans ate Magnoliales leaves.

Results are not significant enough, and there are several botanical issues for supporting this assumption. Pteridophyta are now called differently. Gymnosperms have not only needle-like leaves. APG is not followed. *Machilus nanmu* is not included in Magnoliales anymore. The same is for *Ficus tikoua*. In plants, phytoliths are not only present in leaves. Outlines on phytolith microphotographs are poorly comparable; Does SEM give more information than LM? Are there any leaf fossils of Magnolialean affinity in the same layers? So claims that *Jeholornis* ate leaves and particularly Magnoliales are not definitively demonstrated.

Reviewer #1 (Remarks to the Author):

In their manuscript, Wu et al. report a new specimen of the basally-diverging avialan Jeholornis that preserves dietary gut contents. The authors demonstrate that those contents include phytoliths, a rugged silica-based structure produced by plants. Based on comparison of SEM imagery of those phytoliths to corresponding structures from a sample of extant plants, the authors argue that they were most likely produced by an angiosperm plant (specifically magnolias and relatives). The authors use these results in combination with a GMM analysis of the mandible preserved in the new specimen to argue that angiosperm-based folivory constituted a significant proportion of the diet in Jeholornis.

This study provides a crucial glimpse into the early co-evolution of angiosperm diversity and avialan feeding ecology, and sheds light on the community structure of the extremely important Cretaceous Jehol Biota. Generally, I find the authors' arguments to be reasonable, compelling and well-supported by the presented data. The artistic reconstruction of Jeholornis (Fig. 4) is exceptional – please pass my regards on to the artist. My only major comments are requests for additional information in some areas to better support their interpretations.

1. The authors do not explicitly discuss – and therefore do not reject – any alternative interpretations of the structures they refer to as phytoliths. The comparative imagery the authors provide is quite convincing that they are, but some justification for the rejection of alternative interpretations should be provided. Or, if there are no other structures they might be, that should be stated explicitly.

Reply: Thank you for the nice comments.

First of all, characteristics used to referring these structures as phytoliths included a dull-color under orthogonal polarized light, same as the optical properties of opal under transmitted light microscopy. In terms of morphology, phytolith is readily different from other siliceous substances such as mineral silicon, diatoms and sponge spicules. SEM-EDS analyses demonstrate that the main component present in the identified phytoliths is silicon dioxide, which is also the typical for phytoliths (Supplementary Fig. 3).

2. Related to my previous point, additional comparative information should be provided to the diagnosis of the phytoliths as coming from Magnoliales. Minimally, images of the fossil phytoliths should be added to Supplementary Fig. 4-5 to help readers make comparisons more easily. Also, while I am convinced by the authors' assignments of some pictured phytoliths to Magnoliales (e.g. Fig. 1a-b), I am not convinced by the identification of others

as hair base phytoliths. Specifically, to my (admittedly untrained) eye, the phytoliths pictured in Fig. 1d-e looking nothing like Fig. 1f. I recommend either the authors provide more justification for that assignment or drop that line of reasoning completely. I am inclined to suggest the latter – the authors' arguments for angiosperm-focused folivory in the new specimen is convincing enough without the hair cell element.

Reply: We appreciate this suggestion.

Accordingly, we have revised the Supplementary Figs. 5-6 by adding images of the fossil blocky phytolith to help readers make a quick and clear comparison.

There are two key features used to identify hair base fossil phytoliths. One is the center papillae of hair base, the other is the surrounding radiate lines that originated from the nearby cell walls. By comparing the two key characters in the fossil with that of modern hair base, we confidently refer the fossil phytolith as hair base structure. According to the advice, we revised the Supplementary Fig. 7 to make better justification for hair base phytoliths. As shown in Fig. S7, the highlighted blue circles, is the center papillae of hair base. The red outlines the surrounding cells whose cell walls formed the surrounding radiate lines. These two key characteristics recognized here in the fossil hair base are consistent with modern hair base.

See revised figure S5-S6:

Supplementary Figure 5. Comparison between Blocky phytoliths with ridgeline ornament as proposed the modern candidates to which the fossil might relate to (**a-i**) and Blocky fossil phytoliths extracted from the digestive tract content of *Jeholornis* (IVPP V14978)(**j**) **a**, *Equisetum ramosissimum* from Yunnan; **b**, *Pinus armandii* from Tibet; **c-d**, *Oryza sativa* from Zhejiang; **e**, *Quercus* sp. from Sichuan; **f**, *Magnolia* sp. from Luzon; **g-i**, *Machilus nanmu* from Sichuan; **j**, fossil blocky phytoliths with wavy ridgelines from the stomach content of *Jeholornis* (IVPP V14978), possibly related to blocky phytoliths in modern Magnoliales leaves (f-i).

Supplementary Figure 6. Comparison between Blocky fossil phytoliths extracted from the digestive tract content of *Jeholornis* (IVPP V14978)(**a**) and Blocky Phytoliths with ridgeline ornament extracted from modern Magnoliaceae(**b-h**). **a**, fossil blocky phytoliths with wavy ridgelines from the stomach content of *Jeholornis* (IVPP V14978), possibly related to blocky phytoliths in modern Magnoliales leaves. **b**, *Lirianthe henryi* (Dunn) N.H. Xia & C.Y. Wu; **c**, *Magnolia coco* (Lour.) DC. **d**, *Manglietia decidua* Q.Y. Zheng; **e**, *Manglietia fordiana* Oliv.; **f**, *Yulania*

denudata (Desr.) D.L. Fu; **g**, *Magnolia championii* Benth. **h**, *Yulania amoena* (W.C. Cheng) D.L. Fu.

Supplementary Figure 7. Comparison between hair base phytoliths extracted from the digestive tract content of *Jeholornis* (IVPP V14978) (**d,e**) and hair base phytoliths extracted from extant *Ficus tikoua* leaves (**f**). Blue is center papillae of hair base. Red is the surrounding cells whose cell walls formed the surrounding radiate lines. These two key characteristics from fossil hair base are consistent with modern hair base.

3. I don't find the GMM aspect of this study very useful or relevant – which I suspect is why little attention is paid to it in the manuscript. As the authors note, it seems to do little beyond suggesting a “generalized dietary morphotype,” of which folivory may or may not be a part. The authors’

emphasis on similarity with Hoatzin seems unjustified – indeed, there seem to be several taxa in Fig. 3 that are apparently not herbivorous (they are not colored yellow) but are more closely positioned in parameter space to both Jeholornis and Hoatzin than each other. If the authors decide to keep this section, I think it warrants much more discussion: what are those taxa that are close to Jeholornis or Hoatzin that are not herbivorous? The authors briefly mention that other authors used GMM approaches to reject a granivorous diet for Jeholornis – how are the results of the current paper different from those previous studies – or are they?

Reply:

Thank you for your suggestions. The GMM part is indirect evidence of our study, and rather supplied as side evidence to enrich the new results, which make our conclusion better supported.

Although the phytoliths part is the key, we insistent to keeping GMM part because it could provide us certain supplementary evidence besides the main phytoliths analyses. We modified this part in three aspects according to the precious suggestions

which aim to apply multiple methods in diffent angles to confirm our phytoliths findings.

1) We toned down the importance of GMM outcome, and emphasized it as supplementary analyses to the main phytoliths results throughout the manuscript, by adding texts such as “to provide supplementary test to the phytoliths analyses regarding the leaf consumption of *Jeholornis*” .

2) Due to the complexity regarding avian diets, it is impossible to project every category of known diets in living birds in the PCA plots, otherwise it will be very confusing to the readers. However, we agree with the suggestions raised here which better to project certain other diet groups so that the readers could have comparisons. Thus we adapted the dominant diet information to be four categories related to our topic here now: 1) Plant eaters (Anseriformes), 2) Plant eaters (Galliformes), 3) Fruits and 4) Other diets, and revised Figure 3 to better deliever the results.

The hoatzin is plotted very close to *Jeholornis* along both PC1 and PC2 axes in Figure 3. We also added some texts for the qualitative comparisons now:

“The qualitative comparison of the 3D morphology of mandible between Jeholornis and the hoatzin also provides strong similarities, including the reflexed rostral part of the mandible.” In addition, in the revised figure, it is also obvious that Jeholornis is overall closer to the sampled plant eaters rather than to the superficially fruit eaters (considering that it has been proved to consume fruits), especially along PC1 axis. We agree wjith that there are other

taxa close to *Jeholornis* too, e.g., some raptorial birds (*Micrastur*, *Vultur*), which could be due to the convergent evolution of tearing food items adopted in jaw function for these taxa, therefore, we are not using this result as exclusively support of *Jeholornis* being a folivore, but using it as supplementary evidence for its potential leaf consumption instead. We think it is meaningful to keep it here as related supplementary morphological analysis in addition to the phytoliths data, and hope the change would make the similarity between *Jeholornis* and other generalized herbivorous birds more convincing now in the revised figure (see the original and revised figure).

3) The GMM analysis is a reinvestigation of the previous analyses of mandible including *Jeholornis* and modern birds. The difference is the previous analyses were focusing on testing the seed cracking ability of *Jeholornis*, while here we are focusing on the possibility of *Jeholornis* to be a plant eater, and showing certain qualitative comparisons between *Jeholornis* and the hoatzin as pointed out in 2). They are not conflict or covering with each other, just focusing on different topics. However, we added and modified some texts in the manuscript to make this clearer, such as “The previous interpretation derived from this mandible GMM analysis only focused on excluding the possibility of granivory in *Jeholornis*¹⁶, but our reexamination indicates the potential of a more varied plant-based diet in *Jeholornis*, which provides the supplementary evidence to the results of the phytolith analyses that the leaf consumption is at least part of its diet”.

4. The systematic paleontology section in the supplement should more explicitly justify referral of the specimen to *Jeholornis prima*. I have no doubt that the authors are correct in their referral, but they should discuss not only how the new specimen is like *J. prima* (which they do), but also how they reject other possible taxonomic assignments. Also, the authors should provide some information on the locality from which it was recovered.

See detail in SI file;

We have added detail description and comparison of the fossil in supplementary file (see systematic paleontology) ; and we also specified the location of the new fossil is in Gong-gao village, near Chaoyang county in Liaoning province. We make extensively comparison of the new specimen, with that of other Jeholornis taxa and clarify some of the diagnostic features and indicate its most consistent with morphologies of *J. prima*.

We have also added more CL image (SI figure) to detail the anatomy of the new Jeholornis and distinguish it with other Jeholornis taxa as well.

5. Related to my previous point, the line drawing (Fig. 1a) should provide a more direct, literal interpretation of the preserved elements. As presented, the "sketch"-like approach is attractive, but not very useful for discerning much meaningful anatomy, especially in the context of taxonomic diagnosis. I urge the authors to use a program like Illustrator to use well-defined lines with consistent line weights and to represent as much discernable anatomy as possible.

We re-drew the figure entirely and thanks for the suggestion. See revised figure 1.

6. Based on the data availability statement in the paper, it is not clear to me if the authors have already done this, but I *strongly* urge them to place the digital data associated with this paper (e.g., the CT scan data) on an online repository (e.g., MorphoSource) and provide a link to those data in the final version of the paper.

Thank you for pointing this out! We are actually using a published specimen in the GMM part, so it has been stored in MorphoSource. Thanks to your suggestion, we added the following information to the Data Availability part now:

"All the specimens used in the GMM part have been published before including the key specimen Jeholornis STM 3-816, whose computed tomography scanning slices and segmented STL files has already been available in MorphoSource (<https://www.morphosource.org/projects/0000C1212>)¹⁶."

We also uploaded the CL data (skull and stomach regions) to a open server as well, and the data will be made open as the release of the paper. OSF link will be added here.

Some minor note:

Line 61 – some awkward grammar and typos here – try something like: “from the approximate location of the stomach region”

Adopted

from the approximate location of the stomach region”

Line 68-69 – “...from their predatory close outgroups...” – it is unclear what the authors are describing here – are they referring to a folivorous diet as a derived feature relative to the presumably ancestral carnivorous diet of close relatives?

Yes, we mean folivorous diet in dinosaur evolution is a derived trait, as compared to other theropods which are mainly carnivorous.

Line 69-70 – This should be reworded to make clear that the morphometric analysis being referred to here was done within this same paper.

rewrite

Line 74 – “over one hundred” rather than “over a hundred”

adopted

“over a hundred”

Line 86 – “high” is misspelled

changed

Line 100 – “careful” is misspelled but also I don’t think you need this word here

delete

Line 106 – extra space before period

delete

Line 120 – what do you mean by Pterydophita? I am by no means well versed in plant phylogenetics, but it is my understanding that the grouping referred to by “Pterydophita” is not monophyletic, and so it is an obsolete word. But I may be wrong!

Reply: Yes, we have revised it. We use ferns instead.

Line 185 – “few energy” is awkward phrasing – presumably you mean less energy than an alternative food source. I suggest you make that, as well as what those alternatives are, explicit.

changed

Reviewer #2 (Remarks to the Author):

This is a very interesting paper. It deals with time periods outside of my expertise but I nevertheless can comment on the phytolith record. The phytolith record is well conceived, identified, and interpreted. The methods use to isolate and identify phytoliths from the bird's stomach are sound. Magnoliaceae and other basal angiosperm phytoliths such as Lauraceae are well-studied by others and the authors, and it does appear that the bird was eating members of the Magnoliales, supporting other conceptual and interpretive aspects of the paper. How interesting that the diet of a bird this old and important can be studied in this way.

Other parts of the paper are well-written and easy to understand for those like myself who study later time periods in the past. I don't have any substantive comments for revision. I think the paper should definitely be published.

Dolores Piperno

Reply: Thank you for the agree with our main findings and it is nice to get positive feedback like this.

Reviewer #3 (Remarks to the Author):

The work suggests that basal avialans ate Magnoliales leaves. Results are not significant enough, and there are several botanical issues for supporting this assumption.

Pteridophyta are now called differently.

Reply: Yes, we have revised it. We use ferns instead.

Gymnosperms have not only needle-like leaves.

We only find similar phytoliths in needle-leaved gymnosperms, and current study indicated that other gymnosperms (e.g., Ginkgo and Cycas) are poor phytolith producers. We have revised the text to "from leaves of conifers" to make the phytolith origin clear.

APG is not followed.

Reply: We checked all familial and ordinal concepts and taxonomy used in our manuscript, and changed according to APG IV.

Machilus nanmu is not included in Magnoliales anymore.

Reply: Yes, *Machilus nanmu* is not in the Magnoliales order.

As the sister group of Magnoliales, Laurales may inherit several morphological characters through their most recent common ancestors

with Magnoliales. Since the fossil phytoliths are nearly identical in morphology to the phytoliths of extant Magnoliaceae and Lauraceae (belonging to Magnoliales and Laurales), we conclude that these blocky phytoliths most likely derive from the leaves of magnoliids.

The same is for *Ficus tikoua*.

Reply: Yes, we use it as a comparison to illustrate the what the hairy base morphology looks like. We are not suggesting it belongs to the magnoliids. We use this as a comparison to illustrate what the hairy base morphology looks like, and we did not indicate it belongs to the magnoliids.

Ficus tikoua is used here as an example to explain that this type of hairy base phytoliths is widely distributed among angiosperms. This does not affect our main focus on the fossil types belonging to the magnoliids.

In plants, phytoliths are not only present in leaves.

Reply: Although phytoliths are produced not only in leaves but also in other parts of the plant, it is undeniable that the highest production of silica bodies occurs in the leaves, and they have more distinct diagnosis characteristics (we identified them as Magnoliopsida leaf silica bodies based on their morphology). From the perspective of the animal itself (in this case, our bird fossil), it is less likely to eat parts such as branches, trunks, or roots, so we have eliminated other parts of the plant.

Outlines on phytolith microphotographs are poorly comparable; Does SEM give more information than LM?

Reply: The phytoliths we extracted come from the Cretaceous period and may have undergone some degree of weathering (rough outline) during the burial and diagenesis process, so their contours may not be very clear, which is a distinct nature of the old sample itself. Compared to LM, SEM confirms the morphological features observed in LM, but more importantly, it confirms that the component is silica dioxide, which is the main component of phytoliths. Therefore, by using both SEM and LM methods to confirm what we have extracted are phytoliths.

Are there any leaf fossils of Magnolialean affinity in the same layers? So claims that Jeholornis ate leaves and particularly Magnoliales are not definitively

demonstrated.

Reply: We did not find any phytoliths in the surrounding rocks of the bird fossil, which proves that the phytoliths extracted from the bird's abdomen or stomach region were indeed ingested by the bird rather than randomly mixed with the surrounding sediments. Within the new CL scans, the sampled material was also indicated inbedded within the throacic ribs and belong to in-situ abdomen residue. In the Jehol Biota, fossils of the basal angiosperm-like plant group, *Archaeofructus*, have also been discovered (see Sun Ge 2002)

In addition, palynological records also demonstrate the presence of angiosperm taxa in the Jehol Biota, including *Magnoliapollis*, *Liliacidites*, *Asteropollis*, *Knemapollis*, and *Chloranthus* (see Wang XZ 2000)

The fossils may not necessarily have been buried where the bird consumed the magnoliids leaves, and it is possible that the phytoliths were preserved as high concentrated residues because they were consume and accumulated as a large amount in the bird's abdomen. For the phytoliths in sediments it is either not preserved, or unlikely to be recovered due to the scare number in the matrix. Therefore, the absence of phytoliths in the surrounding rocks cannot be used as evidence to refute our conclusion.

Added references

Sun G 2002 *Archaeofructaceae* - - a new basal angiosperm family

Xianzeng W, Dong R, Yufei W. First discovery of angiospermous pollen from Yixian Formation in western Liaoning. *Acta Geologica Sinica* 74, 265-272 (2000).

REVIEWERS' COMMENTS

Reviewer #1 (Remarks to the Author):

I find the revised manuscript from Wu et al. to be sufficiently improved, and it is suitable for publication save for two minor details.

1. Several spelling errors are present - a thorough copy edit should be done.
2. I understand that the Fig. S7 focuses in on panels d-f from Fig. 2, but the panels should be re-lettered starting with a.

Reviewer #2 (Remarks to the Author):

I agree with the authors that the phytoliths are likely to be from leaves of the Magnoliales. I'm less certain about their identification of the hair base phytoliths. Some weathering of those phytoliths may have taken place as the authors suggest, but I think the final version of the paper would be stronger without the inclusion of what the authors believe are hair base phytoliths.

REVIEWERS' COMMENTS

Reviewer #1 (Remarks to the Author):

I find the revised manuscript from Wu et al. to be sufficiently improved, and it is suitable for publication save for two minor details.

Reply: we appreciate your valuable suggestions and we are glad the manuscript has been greatly improved.

1. Several spelling errors are present - a thorough copy edit should be done.

Reply: Yes, we have checked and revised all.

2. I understand that the Fig. S7 focuses in on panels d-f from Fig. 2, but the panels should be re-lettered starting with a.

Reply: Thank you for the advice and we have revised panels starting with a.

Reviewer #2 (Remarks to the Author):

I agree with the authors that the phytoliths are likely to be from leaves of the Magnoliales. I'm less certain about their identification of the hair base phytoliths. Some weathering of those phytoliths may have taken place as the authors suggest, but I think the final version of the paper would be stronger without the inclusion of what the authors believe are hair base phytoliths.

Reply: Thank you for the suggestion and we have adopted the suggest. We revised the discussion about hair base phytoliths as:

We found another type of fossil phytoliths, which has a round shape with protuberant center and similar to the modern hair base phytoliths.